# IMPROVING ABSTRACTIVE DIALOGUE SUMMARIZATION WITH CONVERSATIONAL STRUCTURE AND FACTUAL KNOWLEDGE

## ABSTRACT

Recently, people have been paying more attention to the abstractive dialogue summarization task. Compared with news text, the information flows of the dialogue exchange between at least two interlocutors, which leads to the necessity of capturing long-distance cross-sentence relations. In addition, the generated summaries commonly suffer from fake facts because the key elements of dialogues often scatter in multiple utterances. However, the existing sequence-to-sequence models are difficult to address these issues. Therefore, it is necessary to explore the implicit conversational structure to ensure the richness and faithfulness of generated contents. In this paper, we present a Knowledge Graph Enhanced Dual-Copy network (KGEDC), a novel framework for abstractive dialogue summarization with conversational structure and factual knowledge. We use a sequence encoder to draw local features and a graph encoder to integrate global features via the sparse relational graph self-attention network, complementing each other. Besides, a dual-copy mechanism is also designed in decoding process to force the generation conditioned on both the source text and extracted factual knowledge. The experimental results show that our method produces significantly higher ROUGE scores than most of the baselines on both SAMSum corpus and Automobile Master corpus. Human judges further evaluate that outputs of our model contain more richer and faithful information.

## 1 INTRODUCTION

Abstractive summarization aims to understand the semantic information of source texts, and generate flexible and concise expressions as summaries, which is more similar to how humans summarize texts. By employing sequence-to-sequence frameworks, some encouraging results have been made in the abstractive summarization of single-speaker documents like news, scientific publications, etc (Rush et al., 2015; See et al., 2017; Gehrmann et al., 2018; Sharma et al., 2019). Recently, with the explosive growth of dialogic texts, abstractive dialogue summarization has begun arousing people's interest. Some previous works have attempted to transfer general neural models, which are designed for abstractive summarization of non-dialogic texts, to deal with abstractive dialogue summarization task (Goo & Chen, 2018; Liu et al., 2019; Gliwa et al., 2019).

Different from news texts, dialogues contain dynamic information exchange flows, which are usually informal, verbose and repetitive, sprinkled with false-starts, backchanneling, reconfirmations, hesitations, and speaker interruptions (Sacks et al., 1974). Furthermore, utterances are often turned from different interlocutors, which leads to topic drifts and lower information density. Therefore, previous methods are not suitable to generate summaries for dialogues. We argue that the conversational structure and factual knowledge are important to generate informative and succinct summaries. While the neural methods achieve impressive levels of output fluency, they also struggle to produce a coherent order of facts for longer texts (Wiseman et al., 2017), and are often unfaithful to input facts, either omitting, repeating, hallucinating or changing facts. Besides, complex events related to the same element often span across multiple utterances, which makes it challenging for sequence-based models to handle utterance-level long-distance dependencies and capture cross-sentence relations.

To mitigate these issues, an intuitive way is to model the relationships between textual units within a conversation discourse using graph structures, which can break the sequential positions of textual units and directly connect the related long-distance contents. In this paper, we present the Knowledge Graph Enhanced Dual-Copy network (KGEDC), a novel network specially designed for abstractive dialogue summarization. A graph encoder is proposed to construct the conversational structure in utterance-level under the assumption that utterances represent nodes and edges are semantic relations between them. Specifically, we devise three types of edge labels: speaker dependency, sequential context dependency, and co-occurring keyword dependency. The edges navigate the model from the core fact to other occurrences of that fact, and explore its interactions with other concepts or facts. The sparse dialogue graph only leverages related utterances and filters out redundant details, retaining the capacity to include concise concepts or events. In order to extract sequential features at token-level, a sequence encoder is also used. These two encoders cooperate to express conversational contents via two different granularities, which can effectively capture long-distance cross-sentence dependencies.

Moreover, considering that the fact fabrication is a serious problem, encoding existing factual knowledge into the summarization system should be an ideal solution to avoid fake generation. To achieve this goal, we firstly apply the OpenIE tool (Angeli et al., 2015) and dependency parser tool (Manning et al., 2014) to extract the factual knowledge in the form of relational tuples: (subject, predicate, object), which construct a knowledge graph. These tuples describe facts and are regarded as the skeletons of dialogues. Next, we design a dual-copy mechanism to copy contents from tokens of the dialogue text and factual knowledge of the knowledge graph in parallel, which would clearly provide the right guidance for summarization.

To verify the effectiveness of KGEDC, we carry out automatic and human evaluations on SAMSum corpus and Automobile Master corpus. The experimental results show that our model yield significantly better ROUGE scores (Lin & Hovy, 2003) than all baselines. Human judges further confirm that KGEDC generates more informative summaries with less unfaithful errors than all models without the knowledge graph.

## 2    RELATED WORK

**Graph-based summarization**    Graph-based approaches have been widely explored in text summarization. Early traditional works make use of inter-sentence cosine similarity to build the connectivity graph like LexRank (Erkan & Radev, 2004) and TextRank (Mihalcea & Tarau, 2004). Some works further propose discourse inter-sentential relationships to build the Approximate Discourse Graph (ADG) (Yasunaga et al., 2017) and Rhetorical Structure Theory (RST) graph (Xu et al., 2019). These methods usually rely on external tools and cause error propagation. To avoid these problems, neural models have been applied to improve summarization techniques. Tan et al. (2017) proposed a graph-based attention mechanism to discover the salient information of a document. Fernandes et al. (2019) developed a framework to extend existing sequence encoders with a graph component to reason about long-distance relationships. Zhong et al. (2019) used a Transformer encoder to create a fully-connected graph that learns relations between pairwise sentences. Nevertheless, the factual knowledge implied in dialogues is largely ignored. Cao et al. (2017) incorporated the fact descriptions as an additional input source text in the attentional sequence-to-sequence framework. Gunel et al. (2019) employed an entity-aware transformer structure to boost the factual correctness, where the entities come from the Wikidata knowledge graph. In this work, we design a graph encoder based on conversational structure, which uses the sparse relational graph self-attention network to obtain the global features of dialogues.

**Abstractive dialogue summarization**    Due to the lack of publicly available resources, the work for dialogue summarization has been rarely studied and it is still in the exploratory stage at present. Some early works benchmarked the abstractive dialogue summarization task using the AMI meeting corpus, which contains a wide range of annotations, including dialogue acts, topic descriptions, etc (Carletta et al., 2005; Mehdad et al., 2014; Banerjee et al., 2015). Goo & Chen (2018) proposed to use the high-level topic descriptions (e.g. costing evaluation of project process) as the gold references and leveraged dialogue act signals in a neural summarization model. They assumed that dialogue acts indicated interactive signals and used these information for a better performance. Customer service interaction is also a common form of dialogue, which contains questions of the user

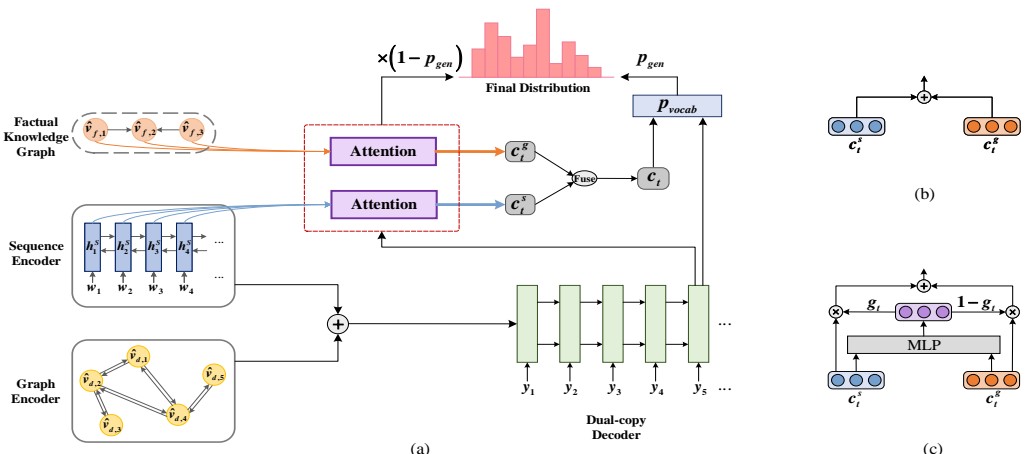

Figure 1: (a) A general architecture of Knowledge Graph Enhanced Dual-copy network for abstractive dialogue summarization. (b) The first approach for fusion, which simply concatenates two context vectors. (c) The second approach for fusion, which uses MLP to build a gate network and combines context vectors with the weighted sum.

and solutions of the agent. Liu et al. (2019) collected a dialogue-summary dataset from the logs in the DiDi customer service center. They proposed a novel Leader-Writer network, which relies on auxiliary key point sequences to ensure the logic and integrity of dialogue summaries and designs a hierarchical decoder. The rules of labeling the key point sequences are given by domain experts, which needs to consume a lot of human effort. For Argumentative Dialogue Summary Corpus, Ganesh & Dingliwal (2019) used the sequence tagging of utterances for identifying the discourse relations of the dialogue and fed these relations into an attention-based pointer network. From consultations between nurses and patients, Liu et al. (2019) arranged a pilot dataset. They presented an architecture that integrates the topic-level attention mechanism in the pointer-generator network, utilizing the hierarchical structure of dialogues. Because the above datasets all have a low number of instances and the quality of them is also low, Gliwa et al. (2019)introduced a middle-scale abstractive dialogue summarization dataset (namely SAMSum) and evaluated a number of generic abstractive summarization methods. Although some progress has been made in abstractive dialogue summarization task, previous methods do not develop specially designed solutions for dialogues and are all dependent on sequence-to-sequence models, which can not handle the utterance-level long-distance dependency, thus causing inaccuracies for rephrased summaries. We present a dual-copy mechanism to directly extract facts from the knowledge graph, which enhances the credibility of the summarization generation.

## 3 METHODOLOGY

In this section, we introduce the Knowledge Graph Enhanced Dual-Copy network, as displayed in Figure 1 (a). Our framework consists of four modules including a sequence encoder, a graph encoder, a factual knowledge graph, and a dual-copy decoder. Importantly, we first present two types of graphs: Dialogue Graph which constructs conversational structures, and Factual Knowledge Graph which directly extracts fact descriptions from source dialogues. Then, the dual-copy decoder generates faithful summaries by embedding the semantics of both source utterances and factual knowledge.

### 3.1 SEQUENCE ENCODER

Considering that the contextual information of dialogues usually flows along the sequence, the sequential aspect of the input text is also rich in meaning. Taking the dialogue $D$ as an example, we feed the tokens of it one-by-one into a single-layer bidirectional LSTM, producing a sequence of encoder hidden states $\{h_1^S, h_2^S, ..., h_n^S\}$. The BiLSTM at the time step $i$ is defined as follows:

$$h_i^S = BiLSTM(w_i, h_{i-1}^S) \tag{1}$$

where $w_i$ is the embedding of the $i$-th token in dialogue, and $h_i^S$ is the concatenation of the hidden state of a forward LSTM and a backward LSTM.

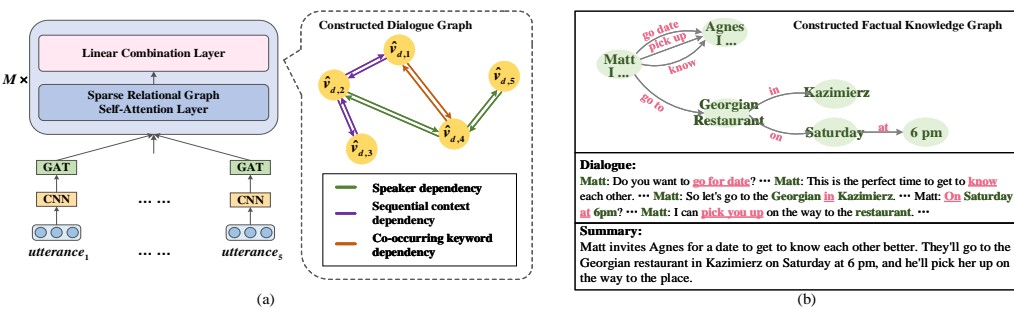

Figure 2: (a) The detailed construction process of a graph encoder (self-loops are omitted for simplification in the constructed dialogue graph). (b) An example of a constructed factual knowledge graph from a dialogue.

## 3.2 GRAPH ENCODER

Given a constructed dialogue graph $G_d = (V_d, E_d)$, $V_d = \{v_{d,1}, ..., v_{d,m}\}$ corresponds to $m$ utterances in the dialogue. $E_d$ is a real-value edge weight matrix and $e_{d,ij} \neq 0$ ($i \in \{1, ..., m\}, j \in \{1, ..., m\}$) indicates that there is an edge between the $i$-th utterance and the $j$-th utterance. We then use graph neural networks to update the representations of all utterances to capture long-distance cross-sentence dependencies, as shown in Figure 2 (a).

### 3.2.1 NODE INITIALIZATION

We first use a Convolutional Neural Network (CNN) with different filter sizes to obtain the representations $x_j$ for the $j$-th utterance $v_{d,j}$. We also do pos tagging on dialogues with off-the-shelf tools such as Stanford CoreNLP (Manning et al., 2014) and select nouns, numerals, adjectives, adverbs, and notional verbs as keywords. Each keyword is transformed into a real-valued vector representation $\varepsilon_i$ by looking up the word embedding matrix, which is initialized by a random process. The attention mechanism is then designed to characterize the strength of contextual correlations between utterances and keywords. The representation of the utterance $\hat{v}_{d,j}$ is updated as a weighted sum of these keyword vector $k_i$ ($i \in \{1, ..., K\}$):

$$A_{ij} = f(\varepsilon_i, x_j) = \varepsilon_i^T x_j$$
$$\alpha_{ij} = softmax_i(A_{ij}) = \frac{exp(A_{ij})}{\sum_{k \in N_j} exp(A_{kj})} \tag{2}$$
$$\hat{v}_{d,j} = \sigma(\sum_{i=1}^{K} \alpha_{ij} W_a \varepsilon_i)$$

where $K$ is the number of keywords in each dialogue, $W_a$ is a trainable weight, and $\alpha_{ij}$ is the attention coefficient between $\varepsilon_i$ and $x_j$.

### 3.2.2 EDGE INITIALIZATION

If we hypothesize that each utterance is contextually dependent on all the other utterances in the dialogue, then a fully connected graph would be constructed. However, this leads to a huge amount of computation. Therefore, we adopt a strategy to construct the edges of the graph, which relies on the various semantic relations among utterances. We define three types of edge labels: speaker dependency, sequential context dependency, co-occurring keyword dependency (A.1).

**Speaker dependency:** The relation depends on where the same speaker appears in the dialogue. In other words, if the utterances belong to the same speaker, we will set an edge between them.

**Sequential context dependency:** The relation describes the sequential utterances that occur within a fixed-size sliding window. In this scenario, each utterance node $v_{d,i}$ has an edge with the immediate $p$ utterances of the past $v_{d,i-1}, v_{d,i-2}, ..., v_{d,i-p}$, and $f$ utterances of the future: $v_{d,i+1}, v_{d,i+2}, ..., v_{d,i+f}$.

**co-occurring keyword dependency:** The relation means that all utterances containing the same keyword are connected.

We further change the existing undirected edges into bidirectional edges and add self-loops to enhance the information flow.

### 3.2.3 ITERATIVE REFINEMENT

To modularize architecture design, multiple blocks, which all consist of the sparse relational graph self-attention layer and the linear combination layer, are stacked to capture long-distance dependencies missed by sequence encoder for better summarization.

**Sparse Relational Graph Self-Attention Layer** Different parts of the dialogue history have distinct levels of importance that may influence the summary generation process. We use a sparse attention mechanism to focus more on the salient utterances. The full self-attention operation in Transformer (Vaswani et al., 2017) captures the interactions between two arbitrary positions of a single sequence. However, our sparse self-attention operation only calculates the similarities between two connected nodes in the graph and masks the irrelevant edges, which reduces the computation amount to a certain extent. In a single sparse graph attention layer, a node in the graph attends over the local information from 1-hop neighbors, which assigns different attention weights to different neighboring edges. These weights can be learned by the model in an end-to-end fashion. Although the sparse graph attention mechanism aggregates the representations of neighborhood nodes along the dependency paths, this process fails to take semantic relations into consideration, which may lose some important dependency information. Intuitively, neighborhood nodes with different dependency relations should have different influences. We extend the sparse graph attention mechanism with the additional edge labels, which can learn better inter-sentence relations. Specifically, we first map the edge labels into vector representations $R$, and then this layer is designed as follows:

$$head_i^l = Attention(QW_i^{Q,l}, KW_i^{K,l}, VW_i^{V,l}, RW_i^{R,l})$$
$$Attention(Q, K, V, R) = softmax(\frac{Q \times K}{\sqrt{d}} + R)(V + R) \tag{3}$$
$$g^l = [head_1^l; ...; head_H^l]W^{o,l}$$

where $W^o$, $W_i^Q$, $W_i^V$, $W_i^K$, and $W_i^R$ are weight matrices, $H$ is the head number, and $d$ is the dimension of utterance node features $\hat{v}_d \in \mathbb{R}^{m \times d}$. In the first block, $Q$, $K$, and $V$ are $\hat{v}_d$. For the following blocks $l$, they are the linear combination layer output vector $z^{l-1} \in \mathbb{R}^{m \times d}$ of block $l-1$.

**Linear Combination Layer** This layer contains two linear combinations with a $ReLU$ activation in between just as Transformer (Vaswani et al., 2017). Formally, the output of the linear transformation layer is defined as:

$$z^l = ReLU(g^l w_1^l + b_1^l)w_2^l + b_2^l \tag{4}$$

where $w_1$, and $w_2$ are weight matrices. $b_1$, and $b_2$ are bias vectors.

After $M$ identical blocks, we take mean pooling over all nodes and obtain the final representation of the dialogue graph, $h^G = \frac{1}{m}\sum_{i=1}^m z_i^M$. We then concatenate the representation of sequence encoder $h_n^S$ and graph encoder $h^G$ to get the initial state of the decoder $s_0 = [h_n^S; h^G]$.

### 3.3 FACTUAL KNOWLEDGE GRAPH CONSTRUCTION

To construct a factual knowledge graph from an input dialogue, we leverage the Open Information Extraction (OpenIE) tools (Angeli et al., 2015) to obtain relation triples. It is worth noting that although OpenIE is able to give a complete description of the entity relations, the relation triples are not always extractable. Hence, we further adopt a dependency parser and dig out some binary tuples from the parse tree of the sentence to supplement the fact descriptions (A.2). Given a factual knowledge graph $G_f = (V_f, E_f)$, nodes $V_f = \{v_{f,1}, ..., v_{f,m}\}$ represent subjects and objects of the triples, and directed edges $E_f$ represent links which pointed from subjects to objects, with predicates as edge labels. We then obtain mentions via the coreference resolution tool and collapse coreferential mentions of the same entity into one node. The node representations $\hat{v}_{f,i}$ are updated by using sparse relational graph self-attention network (Sec 3.2.3). Figure 2 (b) presents an example of a constructed factual knowledge graph from a dialogue.

Table 1: Results in terms of R-1, R-2, and R-L on the SAMSum corpus test set and Automobile Master corpus test set. For our models, we report average scores with the standard deviation.

| Model | SAMSum | | | Automobile Master | | |
|---|---|---|---|---|---|---|
| | R-1 | R-2 | R-L | R-1 | R-2 | R-L |
| Longest-3 | 32.46 | 10.27 | 29.92 | 30.72 | 9.07 | 28.14 |
| Seq2Seq | 21.51 | 10.83 | 20.38 | 25.84 | 13.82 | 25.46 |
| Seq2Seq+Attention | 29.35 | 15.90 | 28.16 | 30.18 | 16.52 | 29.37 |
| Transformer | 36.62 | 11.18 | 33.06 | 36.21 | 11.13 | 34.08 |
| Transformer+Separator | 37.27 | 10.76 | 32.73 | 37.43 | 11.87 | 34.97 |
| LightConv | 33.19 | 11.14 | 30.34 | 34.68 | 12.41 | 31.62 |
| DynamicConv | 33.79 | 11.19 | 30.41 | 34.72 | 12.45 | 31.86 |
| DynamicConv+Separator | 33.69 | 10.88 | 30.93 | 34.41 | 12.38 | 31.22 |
| Pointer Generator | 38.55 | 14.14 | 34.85 | 39.17 | 15.39 | 34.76 |
| Pointer Generator+Separator | 40.88 | 15.28 | 36.63 | 39.23 | 15.42 | 34.53 |
| Fast Abs RL | 40.96 | 17.18 | 39.05 | 39.82 | 15.86 | 36.03 |
| Fast Abs RL Enhanced | 41.95 | 18.06 | 39.23 | 40.13 | 16.17 | 36.42 |
| KGEDC$_c$ | 43.51±0.6 | 19.34±0.3 | 40.57±0.5 | 42.85±1.2 | 18.11±0.9 | 38.02±1.0 |
| KGEDC$_g$ | 43.87±0.4 | 19.66±0.5 | 41.02±0.7 | 43.35±0.9 | 18.23±0.7 | 38.98±1.1 |

## 3.4 DUAL-COPY DECODER

Our decoder is a hybrid between a single-layer unidirectional LSTM and a pointer network. To obtain high faithfulness summaries, we also devise a dual copy mechanism to focus on both the input tokens and the factual knowledge. At each decoding step $t$, we compute a sequence context vector $c_t^s$ with the attention mechanism:

$$c_t^s = \sum_i a_{i,t}^s h_i^s$$
$$a_{i,t}^s = softmax(u_s^T tanh(W_h^s h_i^s + W_s^s s_t + b_{attn}^s))$$

$$(5)$$

where $u_s$, $W_h^s$, $W_s^s$, and $b_{attn}^s$ are learnable parameters. The context vector of the factual knowledge graph $c_t^g$ can be computed similarly. We fuse $c_t^s$ and $c_t^g$ to build the overall context vector $c_t$. We explore two alternative fusion approaches. The first one is called KGEDC$_c$, as shown in Figure 1 (b), which simply concatenates two context vectors:

$$c_t = [c_t^s; c_t^g]$$

$$(6)$$

The other approach is denoted as KGEDC$_g$, as shown in Figure 1 (c), where we also use MLP to build a gate network and combine context vectors with the weighted sum:

$$g_t = MLP(c_t^s, c_t^g)$$
$$c_t = g_t \odot c_t^s + (1 - g_t) \odot c_t^g$$

$$(7)$$

where $\odot$ means the element-wise dot, and MLP stands for a multi-layer perceptron.

Finally, the next token is generated based on the context vector $c_t$, the decoder state $s_t$, and the previous word $y_{t-1}$.

$$P_{vocab} = softmax(U'(U[s_t, c_t] + b) + b')$$
$$P_{gen} = \sigma(w_s^T s_t + w_c^T c_t + w_y^T y_{t-1} + b_{gen})$$
$$P(y_t) = p_{gen} P_{vocab}(y_t) + (1 - p_{gen})(\sum_{i:y_i=y_t} a_{i,t}^s + \sum_{i:y_i=y_t} a_{i,t}^g)$$

$$(8)$$

where $U$, $U'$, $b$, $b'$, $w_s^T$, $w_c^T$, $w_y^T$, and $b_{gen}$ are learnable parameters. $\sigma$ is the sigmoid function.

## 4 DATASET AND EXPERIMENTAL SETUP

### 4.1 DATASET

We perform our experiments on the SAMSum corpus and the Automobile Master corpus, which are both new corpora for dialogue summarization. Table 2 shows some statistics on two datasets. The SAMSum corpus is an English dataset about natural conversations in various scenes of the real-life

(Gliwa et al., 2019). The standard dataset is split into 14732, 818, and 819 examples for training, development, and test. The Automobile Master corpus is from the customer service question and answer scenarios[1]. We use a portion of the corpus that consists of high-quality text data, excluding picture and speech data. It is split into 183460, 1000, and 1000 for training, development, and test. More statistics are in the A.3.

## 4.2 TRAINING DETAILS

We filter stop words and punctuations from the training set to generate a limited vocabulary size of 40k. The dialogues and summaries are truncated to 500, and 50 tokens, and we limit the length of each utterance to 20 tokens. The word embeddings and edge label embeddings are set to 128 and 32, which are both initialized randomly. The dimension for hidden layer units are 256 and 128 for the sequence encoder and the graph encoder, respectively. For factual knowledge graph, the hidden size is set to 128. For the CNN, each feature map is 128 with filter sizes of 3, 4, and 5. The size of sliding window for sequential context dependency in the dialogue graph is set to 1. We use a block number of 2, and the head number of 4 for sparse relational graph self-attention network. At test time, the minimum length of generated summaries is set to 15, and the beam size is 5. For all the models, we train for 30000 iterations using Adam optimizer (Kingma & Ba, 2014) with an initial learning rate of 0.001 and the batch size of 8. For each of our models, we run five times and report the average R-1, R-2, and R-L scores along with the standard deviation. More details can be seen in A.4.

## 4.3 BASELINE METHODS

For both datasets, we compare our proposed method with the following abstractive models: (1) Longest-3; (2) Seq2Seq+Attention (Rush et al., 2015); (3) Transformer (Vaswani et al., 2017); (4) LightConv (Wu et al., 2019); (5) DynamicConv (Wu et al., 2019); (6) Pointer Generator (See et al., 2017); (7) Fast Abs RL (Chen & Bansal, 2018); (8) Fast Abs RL Enhanced (Chen & Bansal, 2018). More details about baselines are shown in A.5.

## 5 RESULTS AND DISCUSSIONS

### 5.1 MAIN RESULTS

**Results on SAMSum Corpus** The results of the baselines and our model on SAMSum corpus are shown in Table 1. We evaluate our models with the standard ROUGE metric, reporting the F1 scores for ROUGE-1, ROUGE-2, and ROUGE-L. Experiments show that $KGEDC_g$ significantly outperforms $KGEDC_c$, and the gate values apparently reflect the relative reliability of dialogues and fact descriptions. By observation, the inclusion of a Separator[2] is advantageous for most models, because it improves the discourse structure. Compared to the best performing model Fast Abs RL Enhanced, the $KGEDC_g$ model obtains 1.92, 1.60, and 1.79 points higher than it for R-1, R-2, and R-L. The sparse relational graph self-attention operation of graph encoder in our model and the extractive method of Fast Abs RL Enhanced model play a similar role in filtering important contents in dialogues. However, our model does not need to use reinforcement learning strategies, which greatly simplifies the training process. Our model also surpasses the pointer generator model without the graph encoder and factual knowledge graph by 2.99, 4.38, and 4.39 points. This demonstrates the benefit of using implicit structures and knowledge to enhance the faithfulness of summaries.

**Results on Automobile Master Corpus** We observe similar trends on Automobile Master corpus as shown in Table 1. Combined with graph encoder and dual-copy decoder, $KGEDC_g$ achieves Rouge-1, Rouge-2, and Rouge-L of 43.35, 18.23, and 38.98, which outperforms the baselines and our $KGEDC_c$ by different margins. Noticeably, unlike SAMSum corpus, The Fast Abs RL Enhanced model has no obvious advantage over other sequence models. This is because the average number of utterances in this dataset is larger and the information is more scattered. Due to the limited computational resource, we do not apply a pre-trained encoder (i.e. BERT) to our model, which

---

[1]This dataset is released by the AI industry application competition of Baidu.

[2]Separator is a special token added artificially, e.g. <EOU> for models using word embeddings, | for models using subword embeddings. The use of it is proposed by Gliwa et al. (2019).

Table 2: Data statistics. A_D, and A_S are the average lengths of dialogues and summaries. A_T, A_P, A_K, and A_F are the average numbers of turns, speakers, keywords, and facts. C_T and C_F mean the proportions of tokens and facts can be found in the summary.

| SAMSum | | | Automobile Master | | |
|---|---|---|---|---|---|
| A_D | 119 | A_S | 23 | A_D | 182 | A_S | 23 |
| A_T | 11 | A_P | 3 | A_T | 11 | A_P | 2 |
| A_K | 33 | A_F | 29 | A_K | 41 | A_F | 37 |
| C_T | 32% | C_F | 44% | C_T | 18% | C_F | 27% |

Table 3: Human Evaluation for the relevance and readability on the test set of SAMSum corpus and Automobile Master corpus.

| Dataset | Model | Relevance | Readability |
|---|---|---|---|
| SAMSum | PGS | 2.36 | 4.25 |
| | FARE | 2.67 | 4.73 |
| | KGEDC$_g$ | 2.97 | 4.91 |
| Automobile Master | PGS | 2.41 | 4.18 |
| | FARE | 2.59 | 4.35 |
| | KGEDC$_g$ | 2.92 | 4.66 |

Table 4: Ablation results w.r.t the sequence encoder, graph encoder, and factual knowledge graph on SAMSum dataset.

| SE | GE | FKG | R-1($\sigma$) | R-2($\sigma$) | R-L($\sigma$) |
|---|---|---|---|---|---|
| ✓ | ✓ | ✓ | 43.87(0.4) | 19.66(0.5) | 41.02(0.7) |
| ✓ | ✗ | ✓ | 39.62(0.6) | 15.05(0.6) | 35.87(0.5) |
| ✗ | ✓ | ✓ | 36.15(0.5) | 12.38(0.8) | 33.79(0.6) |
| ✗ | ✗ | ✓ | 19.04(1.0) | 8.57(0.9) | 17.82(1.2) |
| ✓ | ✓ | ✗ | 42.71(0.5) | 19.15(0.4) | 40.32(0.5) |
| ✓ | ✗ | ✗ | 38.93(0.7) | 14.16(0.6) | 34.88(0.6) |
| ✗ | ✓ | ✗ | 35.65(0.8) | 11.49(0.7) | 32.57(0.4) |
| ✗ | ✗ | ✗ | 17.52(1.2) | 7.90(1.5) | 16.26(1.0) |

Table 5: Ablation results w.r.t the speaker, sequential context, and co-occurring keyword dependency on SAMSum dataset.

| SD | SCD | CKD | R-1($\sigma$) | R-2($\sigma$) | R-L($\sigma$) |
|---|---|---|---|---|---|
| ✓ | ✓ | ✓ | 43.87(0.4) | 19.66(0.5) | 41.02(0.7) |
| ✓ | ✗ | ✓ | 43.05(0.5) | 18.93(0.5) | 40.33(0.6) |
| ✓ | ✓ | ✗ | 42.38(0.7) | 18.16(0.5) | 39.04(0.4) |
| ✗ | ✓ | ✓ | 41.43(0.6) | 17.54(0.3) | 38.26(0.5) |
| ✓ | ✗ | ✗ | 41.05(0.6) | 16.78(0.4) | 37.23(0.4) |
| ✗ | ✗ | ✓ | 40.66(0.5) | 16.13(0.7) | 36.85(0.7) |
| ✗ | ✓ | ✗ | 40.28(0.8) | 15.71(0.9) | 36.49(0.6) |
| ✗ | ✗ | ✗ | 39.62(0.6) | 15.05(0.6) | 35.87(0.5) |

we will regard as our future work. For the sake of fairness, we only compare with models without BERT.

**Human Evaluation** We further conduct a manual evaluation to analyze the relevance and readability of generated summaries. 50 samples are randomly selected from the test set of SAMSum corpus and Automobile Master corpus respectively, and five native or fluent speakers of English are hired as human annotators to rate summaries generated by our KGEDC$_g$, along with outputs by Pointer Generator+Separator (PGS) and Fast Abs RL Enhanced (FARE). After reading the dialogues, each annotator scores each perspective from 1 (worst) to 5 (best). Relevance is a measure of how much faithful information the summary contains, and readability is a measure of how fluent and grammatical the summary is. For relevance, we consider two types of unfaithful errors: (1) deletion or substitution error-mistakenly deleting or substituting subjects, objects, or clauses, and (2) hallucination error-creating content not present in the input. We ask the annotators to label each type as 1 for the existence of errors and 0 otherwise, and give the scores of relevance according to the percentage of errors. From Table 3, we can see that the KGEDC$_g$ obtains better scores, compared to the baselines without the dialogue graph and factual knowledge graph. This indicates the effectiveness of leveraging conversational structures and knowledge graph representation. More details are in A.6.

## 5.2 ABLATION STUDY

We examine the contributions of three main modules, namely, sequence encoder, graph encoder, and factual knowledge graph, using the best-performing KGEDC$_g$ model on SAMSum corpus. The average results of five experiments with standard deviation ($\sigma$) are shown in Table 4. Firstly, we study the effects of two encoders while keeping the FKG module. We only remove one encoder at a time and find that removing either of them leads performance to drop greatly. Especially, the sequence encoder is slightly more important in overall performance, which suggests that the token-level information is indispensable for generating summaries. Removing both of them results in a very poor R-1, R-2, and R-L scores of 19.04%, 8.57%, and 17.82%, which is because the conversational context is not well represented. Next, after getting rid of the FKG module, all models could not keep as competitive as models with the FKG, which suggests the importance of factual knowledge. The framework of the model without GE and FKG resembles the Pointer Generator and their performances are also similar.

Further, we conduct ablation analysis on the edge labels of the dialogue graph, as shown in Table 5. Removal of SCD edges does not significantly affect the performance because the sequence encoder

Table 6: Results in terms of Rouge-1, Rouge-2, and Rouge-L on the XSum corpus test set.

| Model | R-1 | R-2 | R-L |
|---|---|---|---|
| PGS | 30.27 | 9.85 | 23.63 |
| FARE | 32.03 | 11.64 | 26.11 |
| KGEDC$_c$ | 34.71 | 14.38 | 29.56 |
| KGEDC$_g$ | 35.16 | 14.90 | 30.02 |

can replace their impact to some extent. Removing either of SD edges and CKD edges will result in poor performance. Besides, when two of the three edge types are arbitrarily removed, the performance will be greatly reduced, which demonstrates that capturing the long-distance cross sentence dependencies is important for making information flows in dialogues clearer. As we can also see, removing three types of edges completely is equivalent to deleting the graph encoder, which can not aggregate utterance-level features.

## 5.3 GATE FUSION ANALYSIS

As shown in Table 1, KGEDC$_g$ achieves higher ROUGE scores than KGEDC$_c$. Now, we investigate what the gate network (Eq.7) actually learns and what ratio the $c^s$ and $c^f$ are combined. Figure 9 in A.7 shows the changes of the gate values for the development set during training. For SAMSum corpus, at the beginning, the average gate value exceeds 0.5, which suggests the generated content is biased to choose the source dialogue. As the training goes on, our model gradually realizes that the fact descriptions are more reliable, which leads to a consecutive drop of the gate value. Finally, the average gate value $g_t$ is gradually stabilized to 0.422. Notably, the ratio of gate values is $(1 - 0.422)/0.422 \approx 1.37$ and it is extremely close to the ratio of copying proportions $(0.44/0.32 \approx 1.38)$ shown in Table 2, which seems that our model predicts the copy proportion relative accurately and normalizes it as the gate value.

## 5.4 CASE STUDY

Table 8 in A.8 shows an example of dialogue summaries generated by different models. The summary generated by the PGS repeats the same name "Lilly" and only focuses on some pieces of information in the dialogue. FARE model uses both of the extractive and abstractive methods, which makes the generated summary contain both interlocutors' names: Lilly and Gabriel, and obtains some valid keywords, e.g. pasta with salmon and basil. However, FARE combines the subject and object from different parts of a complex sentence, and usually makes mistakes in deciding who performs the action (the subject) and who receives the action (the object). By using various semantic relations, our model constructs a dialogue graph to capture long-distance cross-sentence dependencies and combines the elements of events correctly to some extent. Factual knowledge is also enhanced via the dual-copy mechanism, which makes generated summaries identify more key information. However, there are some deficiencies in logical reasoning for our models. For example, the future tense is not inferred and the logical relationship between "order food" and "for Lilly" is not recognized.

## 5.5 NON-DIALOGIC SETTING

To verify the generalization ability of our method, we run our models on a non-dialogic dataset XSum with more abstractive summaries. In this case, we remove the edges of speaker dependency. As shown in Table 6, the KGEDC$_g$ gets the best scores, which further demonstrates that our method can capture long-distance cross-sentence dependencies and generate more informative and faithful summaries.

## 6 CONCLUSION

We propose a Knowledge Graph Enhanced Dual-Copy network for abstractive dialogue summarization. Our model explores the conversational structure via various semantic relations to construct a dialogue graph, which can capture long-distance cross-sentence dependencies. In tandem with copying from the source dialogue, our dual-copy mechanism utilizes factual knowledge graph to improve generated summaries. The human evaluation further confirms that the KGEDC produces more informative summaries and significantly reduces unfaithful errors.

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

## A APPENDIX

### A.1 EDGE INITIALIZATION FOR DIALOGUE GRAPH

We illustrate the edge initialization process for the dialogue graph and the dialogue content of which is shown in Figure 4. For speaker dependency, Neville, Don, and Wyatt corresponding to (1, 3), (2, 5), and (4, 6), respectively. For sequential context dependency, the size of sliding window is set to 1. For co-occurring keyword dependency, the selected keywords are Neville, remember, date, wedding, Don, and Wyatt, which construct this type edges between (1, 3), (1, 6), (2, 5), (4, 6).

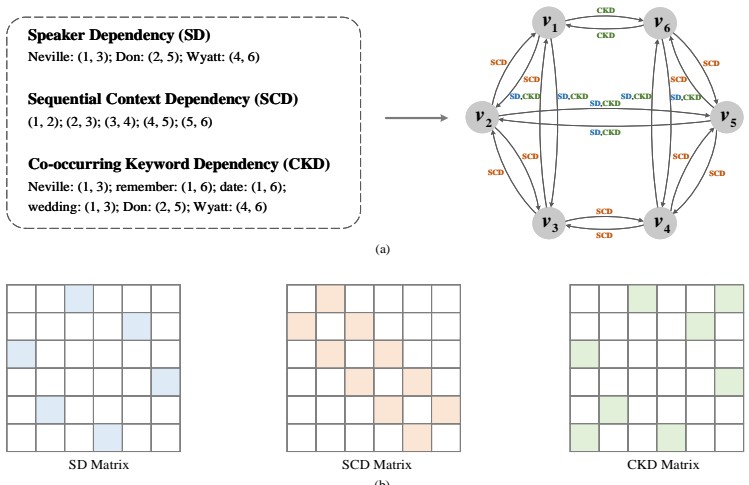

Figure 3: (a) A detailed illustration of the edge initialization for the dialogue graph. The dotted box contains all the ids of utterance pairs for three edge labels. (b) Three types of dependency matric in different colors and white color indicates there is no semantic relation between nodes.

### A.2 FACT TUPLE EXTRACTION

Although OpenIE is able to give a complete description of parts of a fact, the relation triples are not always extractable. In fact, about 18% and 21% of the OpenIE outputs are empty on the SAMSum corpus and Automobile Master corpus, respectively. These empty instances are likely to damage the performance of our model. As observed, each utterance almost contains some binary tuples, while the complete fact tuples are not always available. Therefore, we adopt a strategy to solve this issue. Firstly, we give the part of speech of each token in the utterance via Stanford CoreNLP (Manning et al., 2014). The noun, numeral, adjective, and notional verb, which can express practical significance, are selected as keywords. Then we leverage the dependency parser to dig out the appropriate tuples to supplement the fact descriptions. The dependency parser converts an utterance into the labeled tuples and only the tuples related to keywords are chosen. Finally, we merge the tuples containing the same words, collapse coreferential mentions of the same entity into one word, and order words based on the original sentence to form the fact descriptions. As shown in Figure 4, we give a demonstration of fact tuple extraction in detail. The outputs of OpenIE are empty for some utterances in the dialogue. For 1-th utterance, we filter out the verb-related and noun-related tuples: (anyone, remember), (remember, wedding), and (wedding, date) to form a fact description: anyone remember the wedding date. For 3-th utterance, we filter out the noun-related ,adjective-related, and verb-related tuples: (Tina, mad), (mad, something), (wedding, anniversary), and (have, check) to form a fact description: Tina is mad wedding anniversary. For 6-th utterance, we filter out the noun-related, numeral-related and verb-related tuples: (September, 17), (hope, remember), and (remember, date) to form a fact description: I hope you to remember the date September 17. All these tuples are merged to form facts of the dialogue.

### A.3 STATISTICS OF SAMSUM CORPUS AND AUTOMOBILE MASTER CORPUS

As shown in Figure 5-8, we analyze the length distributions of the dialogue and summary for SAMSum corpus and Automobile Master corpus, respectively.

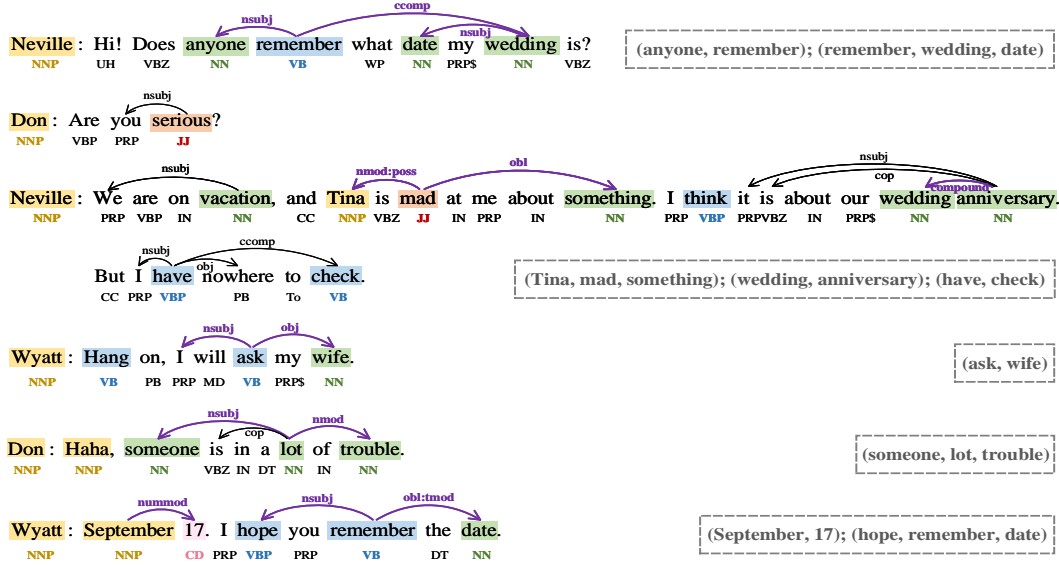

Figure 4: An example of fact tuple extraction in a dialogue. Words with shadings in different colors are selected keywords. The purple arrow denotes the keyword-related tuples and the meaning of the dependency labels can be referred to de Marneffe & Manning (2008). The extracted tuples are shown in the text boxes.

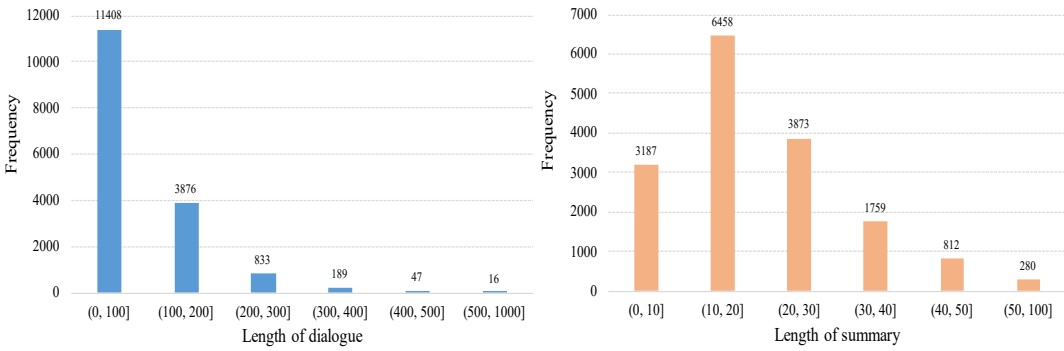

Figure 5: The length distribution of the dialogue for SAMSum corpus.

Figure 6: The length distribution of the summary for SAMSum corpus.

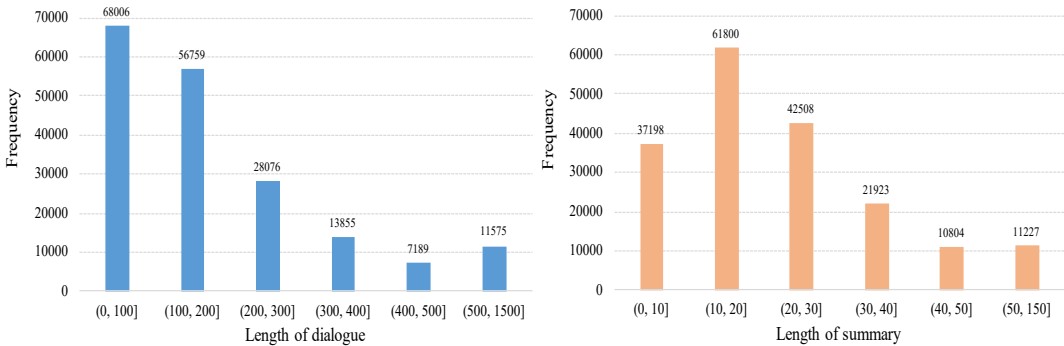

Figure 7: The length distribution of the dialogue for Automobile Master corpus.

Figure 8: The length distribution of the summary for Automobile Master corpus.

## A.4 HYPER-PARAMETER SETTINGS

We tune the hyper-parameters according to results on the development set. We choose the number of blocks $n_b$ from $\{1, 2, 3\}$, and the number of self-attention heads $n_h$ from $\{2, 4, 6\}$. The learning rate $\lambda$ is searched within the range of $\{0.0005, 0.0007, 0.001, 0.005\}$, and the batch size $n_{batch}$ is in the range of $\{8, 16, 32\}$. The settings of hyper-parameters are shown in Table 7.

Table 7: Hyper-parameter settings.

| Parameter | Parameter Name | Value |
|---|---|---|
| $l_d$ | maximum length of dialogue | 500 |
| $l_u$ | maximum length of utterance | 20 |
| $l_{max,s}$ | maximum length of summary | 50 |
| $l_{min,s}$ | minimum length of summary | 15 |
| $d_{word}$ | word embedding size | 128 |
| $d_{edge}$ | edge label embedding size | 32 |
| $d_{s,hidden}$ | hidden embedding size of sequence encoder | 256 |
| $d_{g,hidden}$ | hidden embedding size of graph encoder | 128 |
| $d_{k,hidden}$ | hidden embedding size of factual knowledge graph | 128 |
| $d_{c,hidden}$ | hidden embedding size of CNN | 128 |
| $w_f$ | filter size for CNN | 3-4-5 |
| $w_s$ | sliding window size | 1 |
| $n_b$ | number of blocks | 2 |
| $n_h$ | number of self-attention heads | 4 |
| $n_{beam}$ | beam size | 5 |
| $n_{iter}$ | number of training iteration | 30000 |
| $n_{batch}$ | batch size | 8 |
| $\lambda$ | learning rate | 0.001 |

## A.5 BASELINE DESCRIPTION

In this subsection, we describe baselines in detail.

**Longest-3**: This model is commonly used in the news summarization task, which treats 3 longest utterances in order of length as a summary.

**Seq2Seq+Attention**: This model is proposed by Rush et al. (2015), which uses an attention-based encoder that learns a latent soft alignment over the input text to help inform the summary.

**Transformer**: This model is proposed by Vaswani et al. (2017), which relies entirely on an attention mechanism to draw global dependencies between the input and output.

**LightConv**: This model is proposed by Wu et al. (2019), which has a very small parameter footprint and the kernel does not change over time-steps.

**DynamicConv**: This model is also proposed by Wu et al. (2019), which predicts a different convolution kernel at every time-step and the dynamic weights are a function of the current time-step only rather than the entire context.

**Pointer Generator**: **Fast Abs RL**: This model is proposed by Chen & Bansal (2018), which constructs a hybrid extractive-abstractive architecture, with the policy-based reinforcement learning to bridge together the two networks.

**Fast Abs RL Enhanced**: This model is a variant of Fast Abs RL, which adds the names of all other interlocutors at the end of utterances.

## A.6 DETAILS OF HUMAN ANNOTATIONS

### A.6.1 READABILITY ANNOTATION GUIDELINES

For readability, we make the annotators focus on how fluent and grammatical the summary is and we provide them the following guidelines:

1. First, the annotators judge whether the given sentence is complete or not. If the sentence is incomplete, the annotators rate the score as 1 for it.

2. The annotators can understand the meaning of a complete sentence through their analysis, but there are many grammatical problems in the sentence. The annotators rate the score as 2 or 3 for this sentence.

3. The annotator can easily understand the meaning of the sentence, and there are only minor grammatical problems in it. The annotators rate the score as 4 or 5 for this sentence.

### A.6.2 RELEVANCE ANNOTATION GUIDELINES

For relevance, we make the annotators focus on how much faithful information the summary contains and we consider two types of unfaithful errors: (a) deletion or substitution error-mistakenly deleting or substituting subjects, objects, or clauses, and (b) hallucination error-creating content not present in the input. The guidelines are as follows:

1. Firstly, we ask the annotators to avoid using general knowledge and check whether the given sentence is consistent with the source texts.

2. If the given sentence is faithful to the source texts, the annotators rate the score as 4 or 5.

3. If the given sentence is not faithful to the source texts, the annotators will label each type of unfaithful errors as 1 for the existence of errors and 0 otherwise and give the scores (1, 2, or 3) of relevance according to the percentage of errors.

### A.7 GATE ANALYSIS

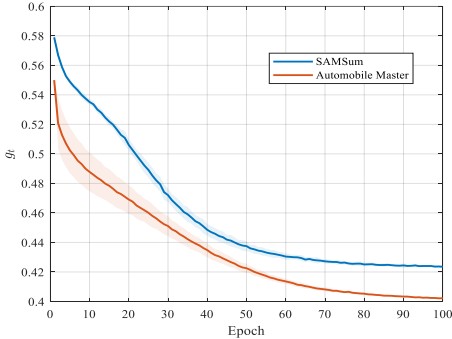

Figure 9: Gates change during training on test set of SAMSum corpus and Automobile Master corpus. Shaded area spans ± std. The blue line represents the result of the SAMSum corpus and the orange line represents the result of the Automobile Master corpus.

A.8   CASE STUDY

Table 8: A case study on the summary generated by different models.

| | |
|---|---|
| **Dialogue** | Lilly: sorry, I'm gonna be late.
Lilly: don't wait for me and order the food.
Gabriel: no problem, shall we also order something for you?
Gabriel: so that you get it as soon as you get to us?
Lilly: good idea!
Lilly: pasta with salmon and basil is always very tasty there. |
| **Reference** | Lilly will be late. Gabriel will order pasta with salmon and basil for her. |
| Longest-3 | Gabriel: no problem, shall we also order something for you? Gabriel: so that you get it as soon as you get to us? Lilly: pasta with salmon and basil is always very tasty there. |
| Pointer Generator Separator (PGS) | Lilly gonna be late. Lilly order pasta. |
| Fast Abs RL Enhanced (FARE) | Gabreil order the food, Lilly thinks pasta with salmon and basil is always very tasty. |
| KGEDC$_g$ | Lilly gonna be late. Gabriel order pasta with salmon and basil. |

