# OpenReview forum: "Improving Abstractive Dialogue Summarization with Conversational Structure and Factual Knowledge"
_ICLR.cc/2021/Conference — Reject_

### Official Review · AnonReviewer4 · 2020-10-28
**Official blind review**

**Rating:** 5
**Confidence:** 4

**Review:**

Summary: This paper proposes a knowledge graph enhanced network to improve abstractive dialog summarization with graphs constructed from the dialog structure and factual knowledge. The dialog graph is composed of utterances as nodes and 3 heuristic types of edges (such as utterances of the same speaker, adjacent utterances). The factual graph is constructed via openIE and dependency parsing, which the authors claim are complementary as the triplets (results of openIE) are not always available.

---
Pros:
+ The proposed method outperforms all the compared baselines on two dialog summarization datasets.
+ Human evaluation shows that the proposed method leads to increased relevance and readability.

---
Cons:
- In the ablation study (Table 3), the performance of each variant is close to the full model, and removing either module still outperforms the compared baselines. Given such performance, I wonder if the difference between the ablated variants and the full model is statistically significant. Also, what is the performance of the proposed method without graph information? Is it effectively the same as a Pointer-Generator?
- Details of human evaluation are lacking. Who are the annotators and how many annotators are there? What is the inter-annotator agreement?
- The paper is poorly represented with unclear descriptions (not only typos/grammar but also definitions of various concepts), which makes it hard to follow. To name a few, in 3.2, the definition of edges gives one the impression that it is a fully connected graph. In 3.2.1, “keyword” is defined after the description of the use of the keyword, and I can’t really tell what is the “keyword neighborhood”.  A general suggestion is to define them beforehand (when they first appear) instead of describing them later or in the footnote.

---
Comments for rebuttal and revised paper

Thanks for providing a detailed response and an improved version of the paper. One thing that I am still concerned with is how come the updated ablation study is so different from the initial results. Originally, the differences between KGEDCg and KGEDCg-GE and KGEDCg-FKG were very minor (one of my questions above), but now the margins are as large as 7+ pts. Given such discrepancies without explanation, I'd hold my original evaluation.

---

> ### Author Response · Authors · 2020-11-22
> **Responses to AnonReviewer4**
>
> Dear reviewer, we thank you for providing very valuable suggestions. We will explain your concerns and answer your questions point by point.
>
> Q1: In the ablation study (Table 3), the performance of each variant is close to the full model, and removing either module still outperforms the compared baselines. Given such performance, I wonder if the difference between the ablated variants and the full model is statistically significant. Also, what is the performance of the proposed method without graph information? Is it effectively the same as a Pointer-Generator?
>
> A1: We further enrich the ablation studies. We do an ablation experiment on the Sequence Encoder (SE), Graph Encoder (GE), and Factual Knowledge Graph (FKG) using the best-performing $\rm KGEDC_g$ model on SAMSum corpus. As we can see, with the GE removed, the R-1, R-2, and R-L scores decrease by 4.25, 4.61, and 5.15 points. With the sequence encoder removed, the R-1, R-2, and R-L scores decrease by 7.72, 7.28, and 7.23 points. With the factual knowledge graph removed, the R-1, R-2, and R-L scores decrease by 1.16, 0.51, and 0.80 points. For all experiments, we run our models five times and report the average scores with the standard deviation, which suggests that the difference between the ablated variants and the full model is statistically significant. Besides, after getting rid of the GE and FKG (i.e. graph information), the R-1, R-2, and R-L scores decrease by 4.94, 5.50, and 6.14 points and. The R-1, R-2, and R-L scores of this ablated variant are 38.93, 14.16, and 34.88, which are similar to the Pointer Generator with 38.55, 14.14, and 34.85 for R-1, R-2, and R-L.
>
> |SE|GE|FKG|R-1 ($\sigma$)|R-2 ($\sigma$)|R-L ($\sigma$)|
> |----|----|-----|------|------|------|
> |✔|✔|✔|43.87(0.4)|19.66(0.5)|41.02(0.7)|
> |✔|✘|✔|39.62(0.6)|15.05(0.6)|35.87(0.5)|
> |✘|✔|✔|36.15(0.5)|12.38(0.8)|33.79(0.6)|
> |✘|✘|✔|19.04(1.0)|8.57(0.9)|17.82(1.2)|
> |✔|✔|✘|42.71(0.5)|19.15(0.4)|40.32(0.5)|
> |✔|✘|✘|38.93(0.7)|14.16(0.6)|34.88(0.6)|
> |✘|✔|✘|35.65(0.8)|11.49(0.7)|32.57(0.4)|
> |✘|✘|✘|17.52(1.2)|7.90(1.5)|16.26(1.0)|
>
> Q2: Details of human evaluation are lacking. Who are the annotators and how many annotators are there? What is the inter-annotator agreement?
>
> A2: (1) We hire five native or fluent speakers of English as human annotators to rate summaries generated by our $\rm KGEDC_g$, along with the outputs by Pointer Generator+Separator (PGS) and Fast Abs RL Enhanced (FARE).
>
> (2) For readability, we make the annotators focus on how fluent and grammatical the summary is and we provide them the following guidelines:
>
> 1. First, the annotators judge whether the given sentence is complete or not. If the sentence is incomplete, the annotators rate the score as 1 for it.
>
> 2. The annotators can understand the meaning of a complete sentence through their analysis, but there are many grammatical problems in the sentence. The annotators rate the score as 2 or 3 for this sentence.
>
> 3. The annotator can easily understand the meaning of the sentence, and there are only minor grammatical problems in it. The annotators rate the score as 4 or 5 for this sentence.
>
> For relevance, we make the annotators focus on how much faithful information the summary contains and we consider two types of unfaithful errors: (a) deletion or substitution error-mistakenly deleting or substituting subjects, objects, or clauses, and (b) hallucination error-creating content not present in the input. The guidelines are as follows:
>
> 1. Firstly, we ask the annotators to avoid using general knowledge and check whether the given sentence is consistent with the source texts.
>
> 2. If the given sentence is faithful to the source texts, the annotators rate the score as 4 or 5.
>
> 3. If the given sentence is not faithful to the source texts, the annotators will label each type of unfaithful errors as 1 for the existence of errors and 0 otherwise, and give the scores (1, 2, or 3) of relevance according to the percentage of errors.
>
> Q3: The paper is poorly represented with unclear descriptions (not only typos/grammar but also definitions of various concepts), which makes it hard to follow. To name a few, in 3.2, the definition of edges gives one the impression that it is a fully connected graph. In 3.2.1, “keyword” is defined after the description of the use of the keyword, and I can’t really tell what is the “keyword neighborhood”. A general suggestion is to define them beforehand (when they first appear) instead of describing them later or in the footnote.
>
> A3: We improve some definitions in our paper, such as the definition of edges given in Sec 3.2 and the definition of keywords in Sec 3.2.1.

---

### Official Review · AnonReviewer3 · 2020-10-28
**Official Blind Review #3**

**Rating:** 6
**Confidence:** 3

**Review:**

What is the paper about, what contributions does it make, what are the main strengths and weakness?

The paper proposes a novel framework, Knowledge Graph Enhanced Dual-Copy network (KGEDC) for abstractive dialogue summarization. Conversational structure and factual knowledge are incorporated in this framework based on graph network to deal with long-distance cross-sentence dependencies and faithfulness respectively. This framework can be decomposed of 1) a sequence encoder to capture contextual information of dialogues flowing along the sequence, 2) a graph encoder via sparse relation graph self-attention network for cross-sentence dependencies, 3) a factual knowledge graph for representing relational tuples extracted from dialogues, 4) a dual-copy decoder to focus on both the input tokens and the factual knowledge. Experimental results on two datasets show the performance gains of the proposed methods over several previous baselines.

STRENGTHS:

1. The motivation is clear and is in line with the model architecture. Two main unresolved problems in abstractive dialogue summarization are consistently concerned in the paper.

2. The experimental results is rather strong and solid. The paper includes the results of different baseline methods for comparison, showing considerable boosts in both automatic and human evaluation metrics.

3. The paper organization and writing is coherent. Besides, the model description is also well detailed.

WEAKNESS:

1. Dialogue graph may not necessarily be sparse. In section 3.2.3, “Sparse Relational Graph Self-Attention Layer” paragraph, the author writes “However, our sparse self-attention operation …, which reduces the quadratic computation to linear”, but no proof is given, actually this seems not to be right. In a dialog where only two persons interchange words in turn, the number of edges for speaker dependency will still be quadratic to turn length.

2. The constructed factual knowledge graph seems to be sparse since the topics various in different dialogues. As shown in figure 2 (b), there are 7 different edge types in one dialogue session. How does the model deal with this issue?

3. Maybe the paper should provide more content on ablation and case study. For example, in equation (3), edge type is considered in attention computation, while the impact for distinguishing 3 different edges is not studied. Moreover, cases, where the proposed method doesn’t perform well, may also be interesting while not included.

4. In figure 1, both Factual Knowledge Graph and Sequence Encoder are used in predicting the next word at the decoder stage, in an attention manner, but why graph encoder is excluded here? Now that utterance representations have been encoded in a graph encoder, attending to these representations may help in doing better decoding.

Typos. Grammar, and Style

1. just beneath equation (3) in section 3.2.3:  w_i^r should be w_i^R.

---

> ### Author Response · Authors · 2020-11-21
> **Responses to AnonReviewer3**
>
> Dear reviewer, we thank you for providing very valuable suggestions. We will explain your concerns and answer your questions point by point.
>
> Q1: Dialogue graph may not necessarily be sparse. In section 3.2.3, “Sparse Relational Graph Self-Attention Layer” paragraph, the author writes “However, our sparse self-attention operation ..., which reduces the quadratic computation to linear”, but no proof is given, actually this seems not to be right.
>
> A1: We make a wrong expression on "which reduces the quadratic computation to linear". In fact, what we want to express is that our sparse self-attention operation reduces the computation amount to a certain extent. In our paper, the constructed dialogue graph is not a complete graph ($|E_d|<{|V_d|^2}$). Therefore, we regard the constructed dialogue graph as a sparse graph.
>
> Q2: The constructed factual knowledge graph seems to be sparse since the topics various in different dialogues. As shown in figure 2 (b), there are 7 different edge types in one dialogue session. How does the model deal with this issue?
>
> A2: We use the sparse relational graph self-attention and mask operation to deal with the factual knowledge graph, where if there is an edge between two nodes, the element of the mask matrix is set to 1, and if there is no edge between two nodes, the element of the mask matrix is set to 0. The edge labels are also encoded into a relational matrix $R$. The attention score is calculated as: $Attention(Q,K,V,R)=softmax(\frac{Q\times K}{\sqrt d}+R)(V+R)$.
>
> Q3: Maybe the paper should provide more content on ablation and case study. For example, in equation (3), edge type is considered in attention computation, while the impact for distinguishing 3 different edges is not studied. Moreover, cases, where the proposed method doesn’t perform well, may also be interesting while not included.
>
> A3: We add an ablation study for three different types of edges. Removal of the Sequential Context Dependency (SCD) edges does not significantly affect the performance (R-`1: 0.82 ($\downarrow$), R-2: 0.73 ($\downarrow$), R-3: 0.69 ($\downarrow$)) because the sequence encoder can replace their impact to some extent. Removing either of the Speaker Dependency (SD) edges and Co-occurring Keyword Dependency (CKD) edges results in poor performance. With the SD edge removed, the R-1, R-2, and R-L scores decrease by 2.44, 2.12, and 2.76 points. With the CKD edge removed, the R-1, R-2, and R-L scores decrease by 1.49, 1.50, and 1.98 points. Besides, when two of the three edge types are arbitrarily removed, the performance will be greatly reduced, which demonstrates that capturing the long-distance cross sentence dependencies is important for making information flows in dialogues clearer. As we can see, removing three types of edges completely is equivalent to deleting the graph encoder, which can not aggregate utterance-level features.
>
> |SD|SCD|CKD|R-1 ($\sigma$)|R-2 ($\sigma$)|R-L ($\sigma$)|
> |----|-----|-----|------|------|------|
> |✔|✔|✔|43.87(0.4)|19.66(0.5)|41.02(0.7)|
> |✔|✘|✔|43.05(0.5)|18.93(0.5)|40.33(0.6)|
> |✔|✔|✘|42.38(0.7)|18.16(0.5)|39.04(0.4)|
> |✘|✔|✔|41.43(0.6)|17.54(0.3)|38.26(0.5)|
> |✔|✘|✘|41.05(0.6)|16.78(0.4)|37.23(0.4)|
> |✘|✘|✔|40.66(0.5)|16.13(0.7)|36.85(0.7)|
> |✘|✔|✘|40.28(0.8)|15.71(0.9)|36.49(0.6)|
> |✘|✘|✘|39.62(0.6)|15.05(0.6)|35.87(0.5)|
>
> In the case study, we add some analysis on the problems of summaries generated by our model. In Table 8 in A.7, the summary generated by our models is "Lilly gonna be late. Gabriel order pasta with salmon and basil". We find that there are some deficiencies in logical reasoning for our models. For example, the future tense is not inferred and the logical relationship between “order food” and “for Lilly” is not recognized.
>
> Q4: In figure1, both Factual Knowledge Graph and Sequence Encoder are used in predicting the next word at decoder stage, in an attention manner, but why graph encoder is excluded here? Now attending to utterance representations encoded in a graph encoder help in doing better decoding.
>
> A4: The representation of graph encoder is used to initialize the initial state of decoder $s_0$, which passes utterance-level representations of dialogues to the decoder. Therefore, representations of graph encoder are used to do better decoding, as shown in $P_{vocab}$. However, in attention manner, we copy source tokens from the sequence encoder and the factual knowledge from the factual knowledge graph as a pointing process, as shown in $\sum_{i:y_i=y_t}a_{i,t}^s+\sum_{i:y_i=y_t}a_{i,t}^g$. In a word, although all of the sequence encoder, graph encoder, and factual knowledge graph are used in predicting the next word at the decoder stage, they are from two aspects. The sequence and graph encoders are used to generate the next token from the vocabulary table, while the sequence encoder and factual knowledge graph are used to generate the next token by copying from the source input and factual knowledge.
>
> We modify the W_i^R in Eq. 3.

---

### Official Review · AnonReviewer1 · 2020-10-28
**Incorporating factual knowledge into dialogue summarization**

**Rating:** 6
**Confidence:** 4

**Review:**

This paper proposes to improve dialogue summarization by encoding the text with a sequential encoder (for token-level contextualization) and a graph encoder (for long-distance and semantic contextualization). A KG is built and considered to be a surrogate for "factual knowledge". A dual-copy mechanism is used while decoding in the hope that direct access to this factual knowledge will enhance the faithfulness of the generated summaries.

The authors use a biLSTM to encoder the utterances. They build a dialogue graph where each utterance is a node and 2 nodes are connected if they have the same speaker, are within a distance d sequentially, or if they have common keywords (nouns, numerals,
adjectives or notional verbs). They also build the KG using OpenIE triples and tuples from the utterance dependency trees. They use a GNN to encode both the utterance graph and the KG and use either concatenation or gated fusion to get the context vector. They copy from both the text sequence and KG during decoding.

They evaluate the method on two datasets: SAMSum corpus, Automobile Master corpus. They get a Rouge-L improvement of ~1.8 and ~2.5 respectively over string baselines.

Pros:
- The method is interesting. Incorporating facts into text generation is an important and interesting area.
- The results look good. They also perform human evaluation which is appreciated.

Cons/Questions:
- It is not clear how the biLSTM encoding and the GE encoding are combined? Is it the same way as done with h^S and h^G?
- I am not satisfied with the ablation study. One important ablation to run would be an experiment without both GE and FKG. Without those two, the model simply becomes (seq2seq + attention). And from Table 1, we see that the R-L for that simple model is 28.16 and 29.37. It is hard for me to convince myself that adding either GE or FKG increases R-L by more than 12 points.
- It would also be interesting to know in what ratio h_G and h_S are combined in the case of gating fusion.
- Since there is already sequential context dependency in GE, what additional advantage does the sequential encoder provide? Why not train only with the graph encoders?

I would consider updating the rating once the authors respond.

---

> ### Author Response · Authors · 2020-11-20
> **Responses to AnonReviewer1**
>
> Dear reviewer, we thank you for providing very valuable suggestions. We will explain your concerns and answer your questions point by point.
>
> Q1: It is not clear how the biLSTM encoding and the GE encoding are combined? Is it the same way as done with $h^S$ and $h^G$?
>
> A1: Maybe I didn't make it clear in the previous manuscript. The $h_n^S$ is the BiLSTM encoding and the $h^G$ is the graph encoding. We concatenate them to get the initial state of decoder $s_0=[h_n^S;h^G]$. Such information is in Sec 3.2.3.
>
> Q2: I am not satisfied with the ablation study. One important ablation to run would be an experiment without both GE and FKG. Without those two, the model simply becomes (seq2seq+attention). And from Table 1, we see that the R-L for that simple model is 28.16 and 29.37. It is hard for me to convince myself that adding either GE or FKG increases R-L by more than 12 points.
>
> A2: We further enrich the ablation studies. We do an ablation experiment on the Sequence Encoder (SE), Graph Encoder (GE), and Factual Knowledge Graph (FKG).  The model without GE and FKG is more similar to the Pointer Generator model instead of a Seq2Seq+Attention model and their performances are similar. The R-1, R-2, and R-L scores of the model without GE and FKG are 38.93, 14.16, and 34.88. The R-1, R-2, and R-L scores of the Pointer Generator are 38.55, 14.14, and 34.85. Compared to $\rm KGEDC_g$,  the R-1, R-2, and R-L scores of the model without GE and FKG decreased by 4.94, 5.50, and 6.14 points.
>
> |SE|GE|FKG|R-1 ($\sigma$)|R-2 ($\sigma$)|R-L ($\sigma$)|
> |----|----|-----|------|------|------|
> |✔|✔|✔|43.87(0.4)|19.66(0.5)|41.02(0.7)|
> |✔|✘|✔|39.62(0.6)|15.05(0.6)|35.87(0.5)|
> |✘|✔|✔|36.15(0.5)|12.38(0.8)|33.79(0.6)|
> |✘|✘|✔|19.04(1.0)|8.57(0.9)|17.82(1.2)|
> |✔|✔|✘|42.71(0.5)|19.15(0.4)|40.32(0.5)|
> |✔|✘|✘|38.93(0.7)|14.16(0.6)|34.88(0.6)|
> |✘|✔|✘|35.65(0.8)|11.49(0.7)|32.57(0.4)|
> |✘|✘|✘|17.52(1.2)|7.90(1.5)|16.26(1.0)|
>
> It is worth noting that, for Seq2Seq+Attention model, the next predicted word entirely depends on $P_{vocab}$ which is a probability distribution over all words in the vocabulary, and $P(y_t)=P_{vocab}$. However, our model both copies words via pointing and generates words from a fixed vocabulary. $p_{gen}$ is used as a soft switch to choose between generating a word from the vocabulary by sampling from $P_{vocab}$, or copying a word from the inputs by sampling from the attention distributions $\sum_{i:y_i=y_t} a_{i,t}^s+\sum_{i:y_i=y_t} a_{i,t}^g$.
>
> Q3: It would also be interesting to know in what ratio $h_G$ and $h_S$ are combined in the case of gating fusion.
>
> A3: Considering there is no $h_G$ and $h_S$ in the paper, only $h^S$ and $h^G$ exist. Besides, we do not use a gate network to combine the $h^S$ and $h^G$. Therefore, we are not sure if you want to know what the ratio $c^s$ and $c^g$ are combined in the case of gating fusion. If we understand it correctly, the following statement will solve your question. We do a gate fusion analysis in Sec 5.3. We record the changes of the gate values $g_t$ for the development set during training in Figure 9 in A.6. For SAMSum dataset, at the beginning, the average gate value exceeds 0.5, which suggests generated contents are biased to choose source dialogues. As the training goes on, our model gradually realizes that fact descriptions are more reliable, which leads to a consecutive drop of the gate value. Finally, the average gate value is gradually stabilized to 0.422. Notably, the ratio of gate values is $(1-0.422)/0.422\approx1.37$ and it is extremely close to the ratio of copying proportion $0.44/0.32\approx1.38$ shown in Table 2 (0.32 and 0.44 mean the proportions of tokens and facts found in the summary), which seems that our model predicts the copy proportion relative accurately and normalizes it as the gate value. For the Automobile Master corpus, the experimental results are consistent with the data distribution.
>
> Q4: Since there is already sequential context dependency in GE, what additional advantage does the sequential encoder provide? Why not train only with the graph encoders?
>
> A4: In Graph Encoder (GE), the sequential context dependency captures the utterance-level contextual information and focuses on changes of speakers’ views due to the continuous utterances. However, the sequence encoder processes the entire dialogue as a sequence and emphasizes the representation of each token in the token-level. The token and utterance granularities are cooperated to express conversational contents. Besides, token representations of the sequence encoder are also involved in the decoding process and the $h_i^s$ is used in the attention mechanism for sequence context vector $c_t^s$, which generates summaries by copying from the source tokens. Therefore, the sequence and graph encoder are both indispensable modules.

---

### Official Review · AnonReviewer2 · 2020-10-30
**Improving Abstractive Dialogue Summarization**

**Rating:** 6
**Confidence:** 4

**Review:**

This paper proposes a new neural pipeline for dialogue summarization that jointly includes word-by-word decoding, an utterance graph, and a factual knowledge graph. The proposed methods is assessed on the SamSUN and  Automobile Master dataset.
The method improves the top reported baselines by 0.5 to 2 points (RED) in both cases.

The paper contains a well-motivated introduction and perform a sound related work section (as far as my judge). They correctly detail their methods in a step-by-step pedagogical procedure. Overall the paper is pretty well-written.

However, I have a few concerns about the experimental section. While this paper's main contribution is a (sensible) mixture of neural blocks, a single page of experiments to evaluate the method is not enough from my perspective. Besides, a 0.5-2 pts increase compared to a Pointer network may be a bit limited in light of the model complexity.
However, my main concern is mostly the lack of analysis of the method. As there are many design choices, this requires multiple ablation studies to validate all of them. Yet, the authors only ablate the factual knowledge graph and graph encoder. Furthermore, the score differences are small, and a few run + std would have been useful to determine whether one change is significant.
Ideas for other ablation studies could be:
 - the choice of edge
 - the attention block
 - removing both graphs
 - fitler-out some specific POS-tagging to see whether some keywords are more important.
It is also sometimes interesting to point out what part of the network may require additional capacity: should we make the bi-LSTM bigger first? The decoder?

Another interesting experiment would be to run the same architecture on a non-dialogue dataset (by naively setting some of the edges). Therefore, it could show whether the architecture can also perform well on non-dialogue long dependency.
Finally, the authors (rightly) complain that there is a lack of abstract summarisation. However, it is still possible to take the Ubuntu dataset, run a baseline + KGDEC, and perform a human evaluation.
I here list some potential experiments, and I am aware that it is unreasonable to perform them all. I mostly want to point out that there is a large spectrum of things that can be done to further demonstrate the validity of their model.

On a second note, some of the neural hyperparameters are missing, such as the convolution kernel/hidden_size to process the vertices.

Overall, the paper has some merits. Easy to read, plenty of implementation details (although some parameters are missing). However, i cannot recommend paper acceptance without at least two or three additional experiments and potentially the error bar.

---

> ### Author Response · Authors · 2020-11-20
> **Responses to AnonReviewer2**
>
> Dear reviewer, thank you for providing very valuable suggestions. We will explain your concerns and answer the point of your questions by point.
>
> Q1: The concern of lacking ablation studies?
>
> A1: We add some ablation experiments to verify the different neural blocks of our model.
> In the first ablation experiment, we explore the contributions of the Sequence Encoder (SE), Graph Encoder (GE), and Factual Knowledge Graph (FKG), using the best-performing $\rm KGEDC_g$ model on SAMSum corpus. Firstly, we study the effects of two encoders while keeping the FKG module. We only remove one encoder at a time and find that removing either of them leads performance to drop greatly. With the graph encoder removed, the R-1, R-2, and R-L scores decrease by 4.25, 4.61, and 5.15 points. With the sequence encoder removed, the R-1, R-2, and R-L scores decrease by 7.72, 7.28, and 7.23 points. Especially, the sequence encoder is slightly more important in overall performance, which suggests that the token-level information is indispensable for generating summaries. Removing both of them results in a very poor R-1, R-2, and R-L scores of 19.04\%, 8.57\%, and 17.82\%, which is because the conversational context is not well represented. Next, after getting rid of the FKG module, all models could not keep as competitive as models with the FKG, which suggests the importance of factual knowledge. In particular, after only  FKG was removed, the R-1, R-2, and R-L scores decrease by 1.16, 0.51, and 0.70 points. The framework of the model without GE and FKG resembles the Pointer Generator and their performances are also similar.
>
> |SE|GE|FKG|R-1 ($\sigma$)|R-2 ($\sigma$)|R-L ($\sigma$)|
> |----|----|-----|------|------|------|
> |✔|✔|✔|43.87(0.4)|19.66(0.5)|41.02(0.7)|
> |✔|✘|✔|39.62(0.6)|15.05(0.6)|35.87(0.5)|
> |✘|✔|✔|36.15(0.5)|12.38(0.8)|33.79(0.6)|
> |✘|✘|✔|19.04(1.0)|8.57(0.9)|17.82(1.2)|
> |✔|✔|✘|42.71(0.5)|19.15(0.4)|40.32(0.5)|
> |✔|✘|✘|38.93(0.7)|14.16(0.6)|34.88(0.6)|
> |✘|✔|✘|35.65(0.8)|11.49(0.7)|32.57(0.4)|
> |✘|✘|✘|17.52(1.2)|7.90(1.5)|16.26(1.0)|
>
> In the second ablation experiment, we explore the effect of the edge labels of the dialogue graph. Removal of the Sequential Context Dependency (SCD) edges does not significantly affect the performance (R-1`: 0.82$\downarrow$, R-2: 0.73$\downarrow$, R-3: 0.69$\downarrow$) because the sequence encoder can replace their impact to some extent. Removing either of Speaker Dependency (SD) edges and Co-occurring Keyword Dependency (CKD) edges results in poor performance. With the SD edges removed, the R-1, R-2, and R-L scores decrease by 2.44, 2.12, and 2.76 points. With the CKD edges removed, the R-1, R-2, and R-L scores decrease by 1.49, 1.50, and 1.98 points. Besides, when two of the three edge types are arbitrarily removed, the performance will be greatly reduced, which demonstrates that capturing the long-distance cross sentence dependencies is important for making information flows in dialogues clearer. As we can see, removing three types of edges completely is equivalent to deleting the graph encoder, which can not aggregate utterance-level features.
>
> |SD|SCD|CKD|R-1 ($\sigma$)|R-2 ($\sigma$)|R-L ($\sigma$)|
> |----|-----|-----|------|------|------|
> |✔|✔|✔|43.87(0.4)|19.66(0.5)|41.02(0.7)|
> |✔|✘ |✔|43.05(0.5)|18.93(0.5)|40.33(0.6)|
> |✔|✔|✘|42.38(0.7)|18.16(0.5)|39.04(0.4)|
> |✘|✔|✔|41.43(0.6)|17.54(0.3)|38.26(0.5)|
> |✔|✘|✘|41.05(0.6)|16.78(0.4)|37.23(0.4)|
> |✘|✘|✔|40.66(0.5)|16.13(0.7)|36.85(0.7)|
> |✘|✔|✘|40.28(0.8)|15.71(0.9)|36.49(0.6)|
> |✘|✘|✘|39.62(0.6)|15.05(0.6)|35.87(0.5)|
>
> We run our models five times and report average scores with the standard deviation.
>
> Q2: Your suggestion for adding experiments on the non-dialogic dataset?
>
> A2: We evaluate our model on a single-document news summarization dataset XSum. In this case, considering that there are no speaker dependencies in news texts, we remove this edge type. Besides, because the XSum dataset is more abstractive and most of the summaries are generated by paraphrasing the input content rather than copying from a single source sentence, we choose the XSum instead of the CNN/Dailymail dataset. The experimental results show that our model can handle long-distance cross-sentence dependencies on various datasets.
>
> |Model|R-1|R-2|R-L|
> |------|-----|-----|------|
> |PGS| 30.27|9.85|23.63|
> |FARE|32.03|11.64|26.11 |
> |$\rm KGEDC_g$|34.71|14.38|29.56|

---

### Decision · Program_Chairs · 2021-01-07
**Final Decision**

**Decision:**

Reject

**Comment:**

The authors address the important task of improving dialogue summarization using conversation structure and factual knowledge.

Pros:
1) Clearly written and well motivated (as acknowledge by all reviewers)
2) Technically sound (the proposed architecture is clearly in line with the problem that the authors are trying to solve)
3) Significant upgrades to the paper after the reviewer comments (in particular the authors have added detailed ablation studies and results on non-dialogue datasets)

Cons:
1) There is a significant difference between the results in the ablation studies in the original version and in the new version. Originally, the differences between KGEDCg and KGEDCg-GE and KGEDCg-FKG were very minor, but now the margins are as large as 7+ pts. I would request the authors to explain this in the final version

The reviewing team felt that while many Qs were sufficiently addressed by the authors, the large difference in the numbers reported for the ablation study in the initial and final version of the paper raises some new Qs which need to be addressed before the paper can be accepted.